# Proteome-wide solubility and thermal stability profiling reveals distinct regulatory roles for ATP

Sindhuja Sridharan [1,2], Nils Kurzawa [1,3], Thilo Werner[2], Ina Günthner[2], Dominic Helm [4], Wolfgang Huber[1], Marcus Bantscheff [2] & Mikhail M. Savitski[1]

Adenosine triphosphate (ATP) plays fundamental roles in cellular biochemistry and was recently discovered to function as a biological hydrotrope. Here, we use mass spectrometry to interrogate ATP-mediated regulation of protein thermal stability and protein solubility on a proteome-wide scale. Thermal proteome profiling reveals high affinity interactions of ATP as a substrate and as an allosteric modulator that has widespread influence on protein complexes and their stability. Further, we develop a strategy for proteome-wide solubility profiling, and discover ATP-dependent solubilization of at least 25% of the insoluble proteome. ATP increases the solubility of positively charged, intrinsically disordered proteins, and their susceptibility for solubilization varies depending on their localization to different membrane-less organelles. Moreover, a few proteins, exhibit an ATP-dependent decrease in solubility, likely reflecting polymer formation. Our data provides a proteome-wide, quantitative insight into how ATP influences protein structure and solubility across the spectrum of physiologically relevant concentrations.

---

[1] Genome Biology Unit, European Molecular Biology Laboratory, 69117 Heidelberg, Germany. [2] Cellzome, A GSK company, 69117 Heidelberg, Germany. [3] Candidate for joint PhD degree from EMBL and Heidelberg University, Faculty of Biosciences, 69120 Heidelberg, Germany. [4] Proteomics Core Facility, European Molecular Biology Laboratory, 69117 Heidelberg, Germany. These authors contributed equally: Sindhuja Sridharan and Nils Kurzawa. These authors jointly supervised this work: Marcus Bantscheff and Mikhail M. Savitski. Correspondence and requests for materials should be addressed to M. B. (email: marcus.x.bantscheff@gsk.com) or to M.M.S. (email: mikhail.savitski@embl.de)

Nucleotide triphosphates (NTPs) regulate numerous biochemical processes in cells as (co-)substrates, allosteric modulators, biosynthetic precursors, and signaling molecules[1–3]. The most abundant NTP in cells, adenosine triphosphate (ATP) has been well studied for its role as an energy source fueling cellular biochemistry as well as a regulatory molecule essential for protein phosphorylation. Besides these canonical roles, ATP has been reported to affect macromolecular assemblies, such as protein complexes[4,5] and membrane-less organelles[6–8]. The second most abundant NTP, guanosine triphosphate (GTP) is well studied for its regulatory roles in cellular signaling and intracellular transport[9]. Recently, both ATP and GTP have been shown to dissolve protein aggregates[10], and have been postulated to function as hydrotropes. However, the lack of system-wide studies to characterize NTP-interactions under conditions approximating the native cellular environment limits our perspective of the diverse physiological roles of NTPs.

Most protein-metabolite interactions are weak and transient, and hence, challenging to be captured on a system-wide scale. Recently, proteome-wide studies that combine a biophysical characteristic of ligand binding with mass spectrometry have improved our understanding of protein-metabolite interactions in extracts of bacteria[4] and mammalian cells[11–14] but have mainly been restricted to the soluble proteome.

Here, we map and quantify proteome-wide NTP-interactions by assessing thermal stability and solubility of proteins in mechanically disrupted cells, which more closely resemble the cellular environment. Our results reveal diverse biological roles of ATP depending on its concentration. We observe ATP specifically interacting with proteins that utilize it as substrate or allosteric modulator at concentrations lower than 500 μM, while it affects protein-protein interactions of protein complexes at mildly higher concentrations (between 1–2 mM). At high concentrations (>2 mM), ATP modulates the solubility state of a quarter of the insoluble proteome, consisting of positively charged, intrinsically disordered, nucleic acid binding proteins, which are part of membrane-less organelles. The extent of solubilization depends on the localization of proteins to different membrane-less organelles. Furthermore, we uncover roles of ATP in regulating protein-DNA interactions of the Barrier to auto-integration factor (BANF1). Our data provide a quantitative proteome-wide map of ATP affecting protein structure and protein complex stability and solubility, providing unique clues on its role in protein phase transitions.

## Results

**Thermal stability maps NTP-protein interaction affinities.** To specifically assess the global roles of ATP and GTP under conditions approximating the native cellular environment, we mechanically disrupted Jurkat cells to obtain crude lysates that retain insoluble proteins, protein condensates, and membrane proteins embedded in lipids[15,16,17]. In these lysates, we studied the effects of ATP and GTP on protein thermal stability on a proteome-wide scale by combining the principle of the cellular thermal shift assay[18] with multiplexed quantitative mass spectrometry[19]. This approach, termed thermal proteome profiling (TPP) (Supplementary Figure 1A), monitors how melting curves of proteins change upon treatment with a single NTP concentration[11,12,13,20] or over a range of concentrations (two dimensional (2D)-TPP) to determine the amounts of NTP required to alter protein thermal stability (Fig. 1a)[14].

Comparison of TPP experiments performed at 2 mM ATP or vehicle revealed similar melting point shifts in crude and gel-filtered lysates (Supplementary Figure 1B, Supplementary Data 1) confirming that crude lysate is a suitable system for detecting

protein-metabolite interactions. Intracellular ATP concentrations are highly variable in cells (1–10 mM)[21], and GTP levels are approximately five-fold lower than ATP[22]. We found that melting point shifts induced by 2 mM ATP and 0.5 mM GTP in crude lysate were selective for known ATP and GTP-binding proteins, respectively (Supplementary Figure 1C, Supplementary Data 2), and were used as maximum concentrations in dose-dependent studies by 2D-TPP (Fig. 1a–c). The 2D-TPP analysis uncovered 753 proteins with increased thermal stability in the presence of ATP, out of the 7549 identified proteins at a 1% false discovery rate (FDR) (Supplementary Figure 2, see methods). We determined the affinity of NTPs to a protein, as the half-maximal effective concentrations of protein stabilization ($EC_{50}$, $pEC_{50} = -\log_{10}(EC_{50})$). The largest and most potently affected group of proteins that were stabilized by ATP represented annotated ATP-binding proteins (315 proteins), validating our approach. A small subset of GTP-binding proteins also showed increased thermal stability with added ATP (55 proteins) (Fig. 1d and Supplementary Data 3). Moreover, ATP reduced the thermal stability of 151 proteins, which were enriched for RNA binding proteins, including ribosomal proteins ($p$-value < 0.001, hypergeometric test), in line with previous observations in *E. coli*[4].

The 2D-TPP experiment with GTP in crude lysate revealed increased thermal stability for 256 out of 7618 identified proteins, with the two largest groups being GTP- (102 proteins) and ATP- (103 proteins) binding proteins (Fig. 1e, Supplementary Data 3), whereas nine proteins showed lower thermal stability. Proteins containing phosphate binding-loops (P-loops) were stabilized by ATP or GTP, as expected[23]. We found that the residues flanking the Walker-A motif have a substantially more constrained composition in GTP-stabilized proteins compared to ATP-stabilized ones (Supplementary Figure 3A–D). Proteins that were stabilized by both NTPs included a small subset of kinases and proteins with P-loops (AAA-type ATPases and small–GTPases) (Supplementary Figure 3E–H, Supplementary Data 3 and S4). Among these were several proteins known to interact with both metabolites, such as Uridine-cytidine kinase (UCK1 and UCK2), casein kinase (CSNK1A1, CSNK1D, CSNK2A1), and succinyl CoA synthetase (SUCLA2 and SUCLG2)[24–27,28]. Importantly, we identified proteins not previously reported to have this property, such as ATP-binding cassette super family proteins (ABCE1, ABCF1), Caseinolytic peptidase B protein homolog (CLPB), Tyrosine-protein kinase (CSK), RuvB-like ATPase (RUVBL2), Rab-GTPases (RAB21, RAB10, RAB2A), Ras-related protein R-Ras2 (RRAS2), and phosphofructokinase (PFKL) (Fig. 1b, c, Supplementary Figure 3I, J, Supplementary Data 3). In summary, we observed crosstalk between ATP- and GTP-induced thermal stabilization, confirming previously known ATP/GTP-binding proteins and uncovering potential new ones, and showed that ATP- and GTP-binding preferences reflect different sequence constraints on the Walker-A motif flanking residues.

Similar to a recent system-wide study in *E. coli*[4], more than half of the ATP-affected proteins were not annotated in UniProt as ATP or GTP binders. These proteins required higher ATP concentrations for stabilization and were enriched for NAD/FAD, DNA, and RNA binding proteins, as well as for components of annotated complexes and regulatory subunits (Fig. 1d). We observed thermal stabilization of several NADH dehydrogenases, such as GAPDH, IDH1, IDH2, MDH1, MDH2, and PDHX, which have been reported to be inhibited by ATP[29–31,32]. This may suggest that higher ATP levels in cells could cause feedback inhibition of these enzymes. Looking specifically at complexes, we observed preferential stabilization of non-ATP-binding subunits that form complexes with at least one ATP-binding subunit ($p$-value < 0.001, Fisher's exact test). This suggests that thermal stabilization of an ATP-binding complex subunit can propagate

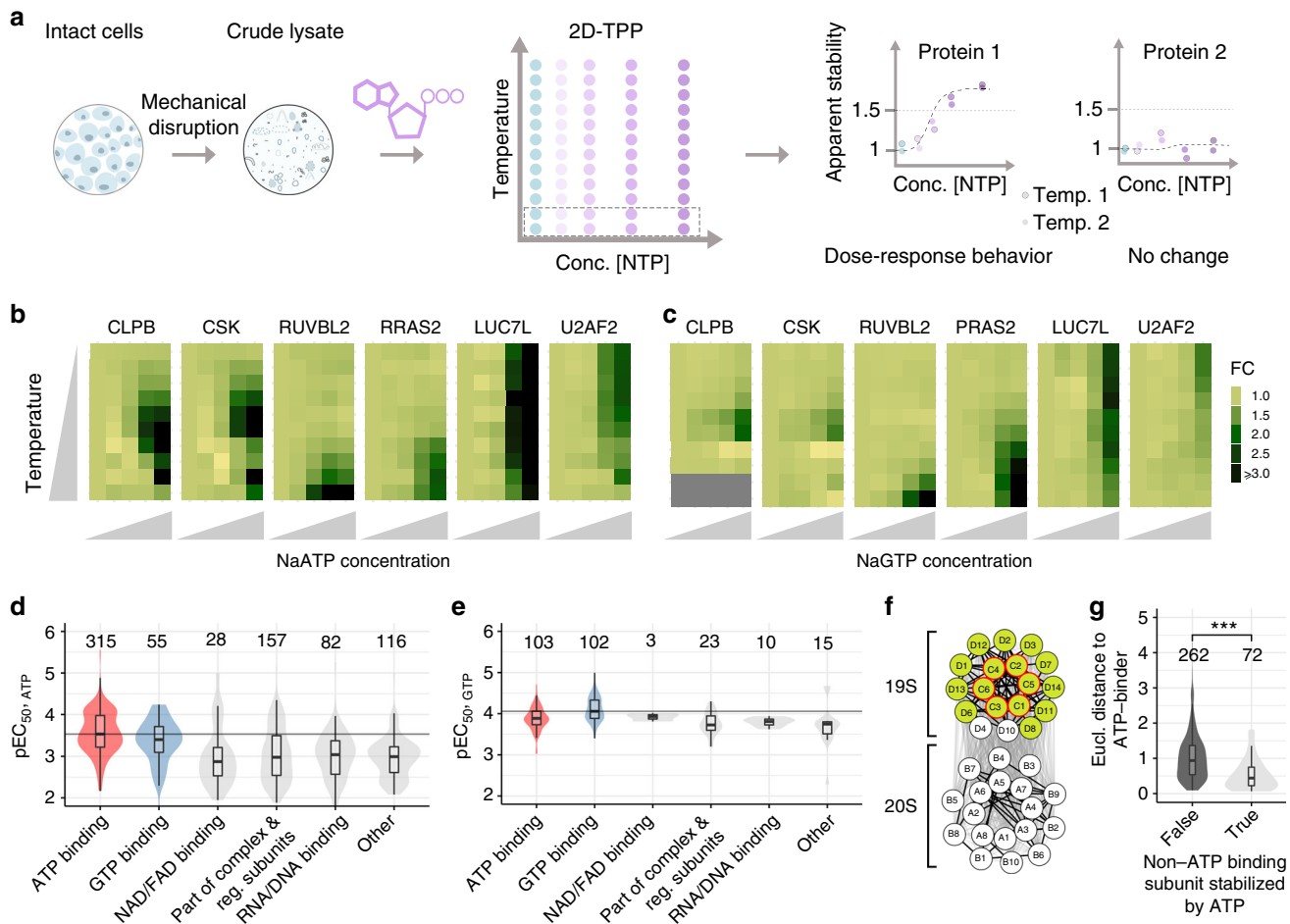

**Fig. 1** Effect of ATP and GTP on proteome thermal stability. **a** Experimental setup of 2D thermal proteome profiling (2D-TPP) using crude lysate system. Dotted rectangular box corresponds to one TMT10 experiment. Blue circles indicate untreated lysate and circle in increasing intensities of purple represent increasing concentration of ligand of interest added to the lysate. Data from three independent experiments have been analyzed. **b** Heat maps showing relative fold changes (FC) of protein abundance upon treatment with ATP (0.005, 0.05, 0.5, and 2 mM) compared to untreated crude lysate (first column on each plot) with increasing temperature (y-axis: 42, 44.1, 46.2, 48.1, 50.4, 51.9, 54.0, 56.1, 58.2, and 60.1 °C). **c** Heat maps showing relative FC of protein abundances upon treatment with GTP (0.001, 0.01, 0.1, and 0.5 mM) compared to untreated crude lysate (first column on each plot) with increasing temperature (y-axis: 42, 44.1, 46.2, 48.1, 50.4, 51.9, 54.0, 56.1, 58.2, 60.1, 62.4, and 63.9 °C). **d, e** Distribution of −log10 half-maximal effective concentration (pEC50) values of different classes of proteins as annotated in UniProt stabilized by ATP (**d**), and GTP (**e**). Black line represents the median pEC50 of annotated ATP- and GTP-binding proteins upon addition of ATP and GTP, respectively. Numbers above violin plots represent number of proteins. **f** Network diagram showing effect of ATP on proteasome stability. Nodes with green (filled) circles indicate ATP-stabilized subunits, and a red outline for nodes show known ATP-binding proteins of the complex. Edge thickness represents the Euclidean distance between the melting profiles of the different subunits. Thick lines are indicative of Euclidean distance less than 0.02. **g** Comparison of Euclidean distances between melting profiles of stabilized ATP-binding complex subunits and non-stabilized non-ATP-binding subunits, (left), and between stabilized ATP-binding complex subunits and stabilized non-ATP-binding subunits (right) within different complexes. Significance levels obtained from a Wilcoxon signed-rank test were encoded as ***p < 0.001. Numbers above violin plots represent numbers of non-ATP-binding subunits in protein complexes with at least one ATP-binding subunit. Violin plots represent relative densities. Center line in all box plots is the median, the bounds of the boxes are the 75 and 25% percentiles i.e., the interquartile range (IQR) and the whiskers correspond to the highest or lowest respective value or if the lowest or highest value is an outlier (greater than 1.5 * IQR from the bounds of the boxes) it is exactly 1.5 * IQR. Source data are provided as a Source Data file for panels B–G

to proximal non-ATP-binding subunits. The proteasome provides a case-in-point: the ATP-binding PSMC subunits and the majority of the non-ATP-binding PSMD subunits of the 19 S regulatory particle were stabilized by ATP. In contrast, the thermal stability of the 20 S core particle, which is devoid of ATP-binding subunits, was unaltered. Comparing the melting curves of proteins by calculating the average Euclidean distance[33], we observed that both 19S and 20S subunits exhibit significant co-melting behaviors (p < 0.001, randomization test) that were different between the particles, indicating that the ATP-induced stabilization propagates only to subunits that are in physical

proximity (Fig. 1f). Global analysis revealed that similar melting behaviors of the stabilized non-ATP-binding subunits and the stabilized ATP-binding subunits within protein complexes is a general characteristic of ATP-induced complex stabilization (Fig. 1g and Supplementary Figure 4).

**ATP depletion in cells recapitulates the ATP-interactome.** To investigate whether our observations made in crude lysates are relevant in intact cells, we reduced ATP levels in Jurkat cells by inhibiting glycolysis and oxidative phosphorylation, using 2-deoxyglucose and Antimycin-A, respectively (Supplementary

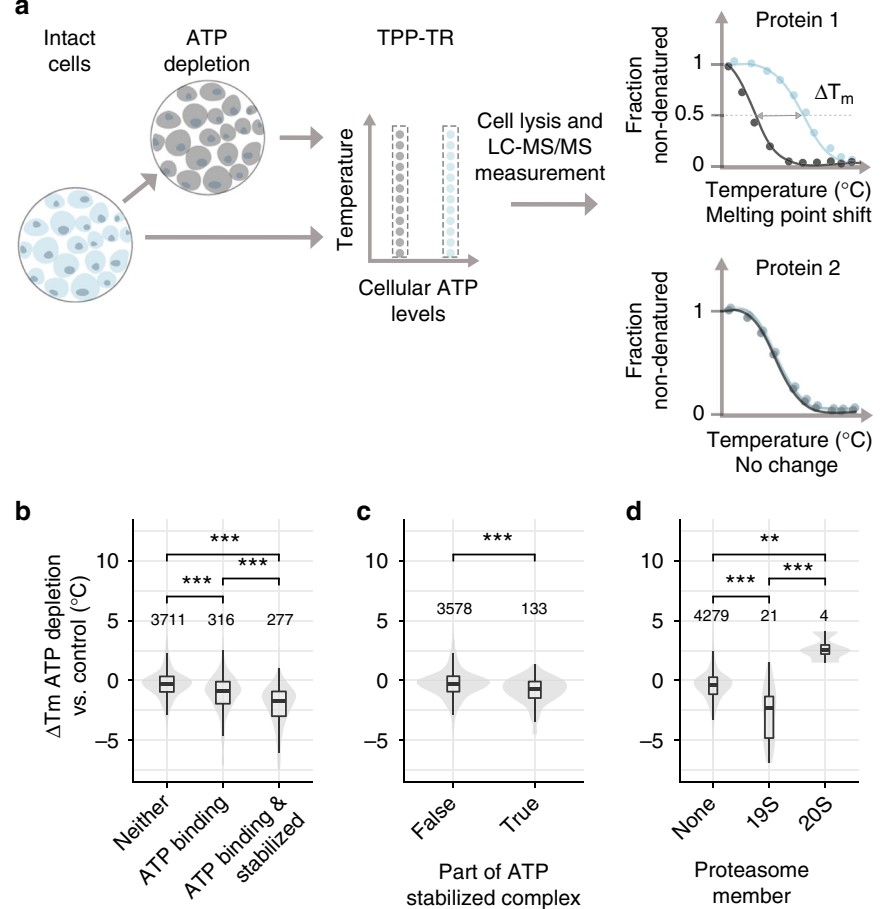

**Fig. 2** Effect of ATP depletion of proteome thermal stability. **a** Experimental setup of TPP on untreated cell (indicated in blue) and cells depleted of ATP by inhibiting glycolysis and oxidative phosphorylation with a combination of 10 mM 2-deoxyglucose and 1 nM Antimycin-A (indicated in grey). Dotted rectangular box corresponds to one TMT10 experiment. **b** Changes in median melting points of non-ATP-binding proteins, ATP-binding proteins, and ATP-binding proteins that were stabilized by ATP in the crude lysate experiment upon ATP depletion in cells from three independent experiments. Significance levels obtained from a Wilcoxon signed-rank test were encoded as *$p < 0.05$, **$p < 0.01$, and ***$p < 0.001$. **c** Changes in melting points of protein complex subunits stabilized by ATP in the crude lysate experiment (right), and all other proteins (left) upon ATP depletion in cells. Significance levels obtained from a Wilcoxon signed-rank test were encoded as *$p < 0.05$, **$p < 0.01$, and ***$p < 0.001$. **d** Changes in melting points of 19S and 20S proteasome subunits, and all other proteins upon ATP depletion in cells. Significance levels obtained from a Wilcoxon signed-rank test were encoded as *$p < 0.05$, **$p < 0.01$, and ***$p < 0.001$. Violin plots represent relative densities. For all box plots, the center line is the median, bounds of the boxes are the 75 and 25% percentiles (i.e., the IQR), and whiskers correspond to the highest or lowest respective value. If the lowest or highest value is an outlier (greater than 1.5*IQR from the bounds of the boxes) whiskers are exactly 1.5*IQR. Numbers above violin plots represent number of proteins. Source data are provided as a Source Data file for panels B–D

Figure 5A, B)[34] and compared the thermal stability profiles of untreated and ATP-depleted cells by TPP (Fig. 2a). The treatment of Jurkat cells with 1 nM Antimycin and 10 mM 2-deoxyglucose resulted in cellular ATP levels being lowered to ~10% of that of untreated cells (Supplementary Figure 5B). Since such depletion of cellular ATP is expected to affect several homeostatic processes, our analysis primarily focuses on assessing the changes in thermal stability of proteins that exhibited changes in the crude lysate system. Indeed, among the 5199 proteins quantified, the ATP-binding proteins (279) that were stabilized by addition of ATP to crude lysates showed the most pronounced thermal destabilization in ATP-depleted cells (Fig. 2b, Supplementary Data 4). Likewise, protein complex subunits that were stabilized by ATP in lysate were destabilized in ATP-depleted cells (Fig. 2c, Supplementary Data 4). Specifically, taking the 26S proteasome as an example, the 19S regulatory particle showed a pronounced destabilization upon ATP depletion compared to the 20S sub-complex, confirming the propagation of thermal stability to

complex members based on their proximity to ligand binding subunits (Fig. 2d). Finally, the proteins that were destabilized by ATP in lysates showed significant thermal stabilization upon cellular ATP depletion ($p$-value < 0.001, Wilcoxon signed-rank test) (Supplementary Figure 5C, Supplementary Data 4). Thus, we conclude that our findings with crude lysates reflect physiologically relevant roles of ATP.

**ATP solubilizes a part of the insoluble proteome**. Crude lysate 2D-TPP experiments revealed several proteins that apparently increase in abundance with higher ATP and GTP concentrations, already at 42 °C (e.g., LUC7L, Fig. 1b, c). This is indicative of increased solubility rather than thermal stability[14], and supports the recently discovered function of ATP as a biological hydro-trope, which solubilizes hydrophobic molecules in aqueous solution[10]. To systematically assess the role of ATP in proteome solubility, we devised an experimental strategy that we termed

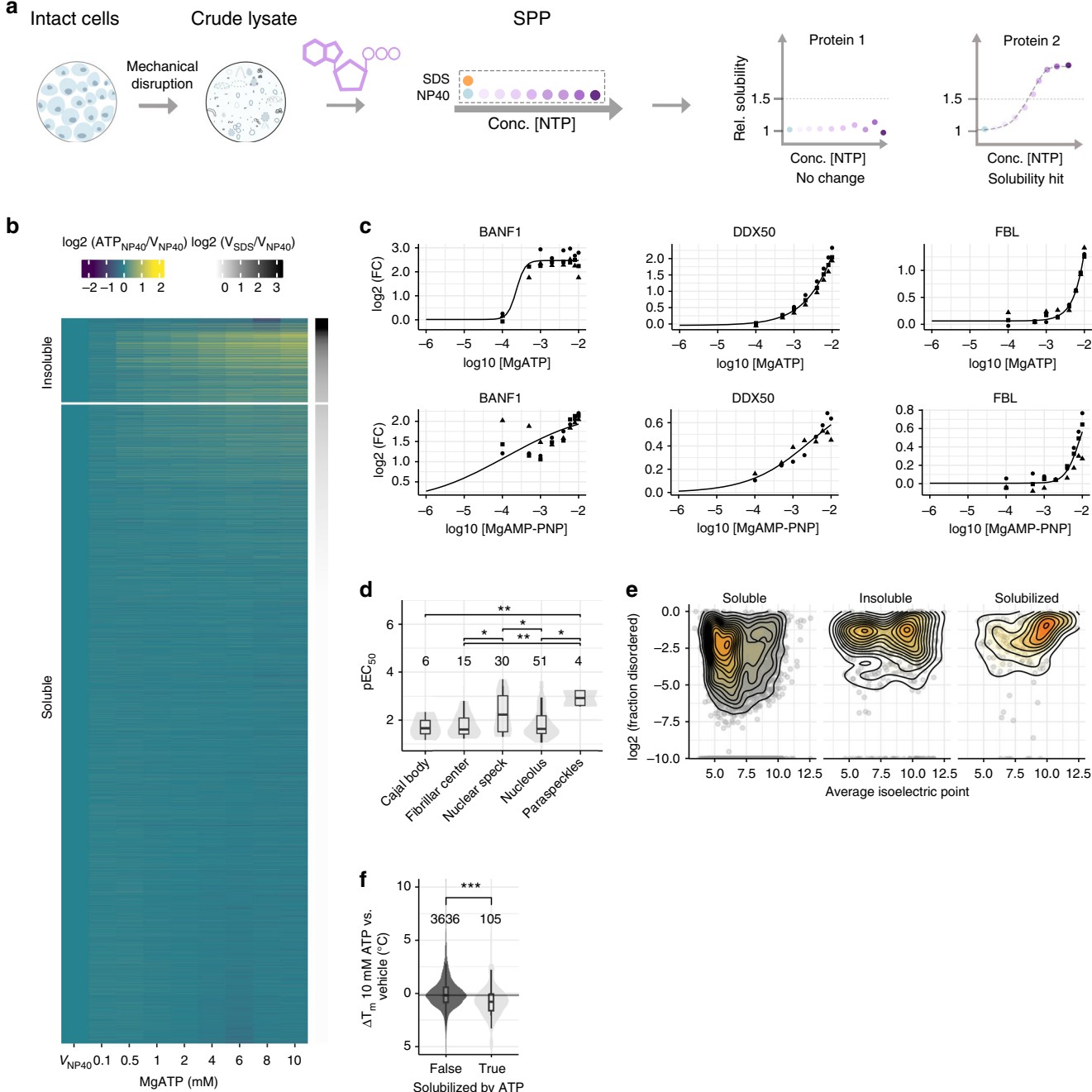

Solubility Proteome Profiling (SPP), which uses multiplexed mass spectrometric analysis to quantify on a proteome-wide scale the concentration-dependent effect of a molecule (e.g., ATP) on the soluble and insoluble populations of individual proteins (Fig. 3a). We switched from Na-ATP to Mg-ATP for SPP experiments since Na-ATP can to some extent deplete endogenous $Mg^{2+}$ levels and thus potentially affect apparent protein solubility. Briefly, crude cell lysate was divided into 10 aliquots, of which nine were treated with either vehicle or increasing concentrations of ATP (0.1–10 mM), followed by solubilization of membrane- or organelle-bound proteins with the mild detergent NP40. A tenth, vehicle-treated aliquot was solubilized with the strong detergent SDS (1%). The ratios between the SDS and NP40 solubilized samples determined for each protein inform on the fractions that exists in an insoluble vs. a soluble state, where a high ratio suggests a higher insoluble fraction[35]. By determining the

half-maximal solubilization concentrations ($EC_{50,s}$), our experiments reveal the differential susceptibility of proteins to solubilization by ATP.

To investigate how the role of ATP as a biological hydrotrope affects the solubility of individual proteins, we examined the insoluble proteome. Proteins that were at least 50% more abundant in the SDS-processed vehicle condition compared to the NP40-processed vehicle condition were defined as part of the insoluble proteome[35]. It should be noted that most of these proteins have a soluble subpopulation (Fig. 3b, Supplementary Data 5). We observed ATP-dependent solubilization for 188 proteins out of the 760 proteins that were identified as insoluble in our experiments, and calculated $EC_{50,s}$ for the dose-dependent solubilization of these proteins (Fig. 3b, Supplementary Data 5). These data showed that ATP solubilized almost 25% of the insoluble proteome. Intriguingly, only 23% of the solubilized

**Fig. 3** Solubility proteome profiling characterizes broad solubilizing effects of ATP on the proteome. **a** Experimental setup of solubility proteome profiling (SPP). Cells were lysed by mechanical disruption and the resulting crude lysate was divided into ten aliquots. Two crude lysate aliquots were treated with vehicle and eight aliquots with different concentrations of a molecule (e.g., ATP—indicated in purple). One vehicle-treated aliquot and eight molecule treated samples were solubilized with a mild detergent (NP40—indicated in blue) and the other vehicle control was solubilized with strong detergent (SDS—indicated in orange). Dotted rectangular box corresponds to one TMT10 experiment. Proteins that showed at least 50% increase in abundance in NP40-processed molecule treatment (at least in one concentration) compared to NP40-processed vehicle, were fitted with sigmoidal dose-response curves and $pEC_{50,s}$ values were calculated. **b** Heat map representation of proteome solubility from three independent experiments. The grey scale represents the $\log_2$ ratio between NP40 and SDS-processed vehicle condition. The color scale represents the $\log_2$ ratio between NP40-processed ATP treated and vehicle-treated conditions. **c** Solubility profiles of BANF1, DDX50, and FBL, following ATP (upper panel) and non-hydrolyzable ATP (AMP-PNP) (lower panel) treatment from three independent experiments, y-axis represents the $\log_2$ ratio of NP40-processed ATP or AMP-PNP treated and vehicle-treated conditions. **d** Distribution of $-\log_{10}$ half-maximal effective concentration ($pEC_{50,s}$) values of ATP-solubilized proteins allocated to different membrane-less organelles. Significance levels obtained from a Wilcoxon signed-rank test were encoded as $*p < 0.05$, $**p < 0.01$, and $***p < 0.001$. **e** 2D density contours of $\log_2$ fraction disorder vs. average isoelectric points of soluble and insoluble proteins, and of those solubilized by ATP. **f** Melting point differences of proteins solubilized by ATP (from SPP data) compared to all other proteins measured by TPP between 10 mM ATP and vehicle-treated crude lysate from two independent experiments. Significance levels obtained from a Wilcoxon signed-rank test were encoded as $***p < 0.001$. Violin plots represent relative densities. For all box plots, the center line is the median, bounds of the boxes are the 75 and 25% percentiles (i.e., the IQR), and whiskers correspond to the highest or lowest respective value. If the lowest or highest value is an outlier (greater than 1.5*IQR from the bounds of the boxes) whiskers are exactly 1.5*IQR. Numbers above violin plots represent number of proteins. Source data are provided as a Source Data file for panels B–F

proteins were annotated as ATP binders while the majority (54%) were annotated as part of membrane-less organelles (Supplementary Figure 6A, Supplementary Data 5). Further, we found that proteins exhibited differential susceptibility for solubilization. For example, proteins such as myosin (MYO1G), which is known to dissociate from actin in the presence of ATP[36], BANF1, and DDX50, were potently solubilized by ATP with sub-micromolar $EC_{50,s}$, whereas other proteins, such as FBL, NOP56, and EIF3H, were only solubilized at much higher ATP concentrations, (Fig. 3c, Supplementary Figure 6B, and Supplementary Data 5). Strikingly, the susceptibility of proteins for being solubilized by ATP depended substantially on their localization in different membrane-less organelles. For example, proteins annotated as part of nuclear speckles solubilized, on average, at lower ATP concentrations than proteins annotated as part of the nucleolus (Fig. 3d). In general, the ATP-solubilized proteins were significantly enriched in disordered regions and had significantly higher isoelectric points than the rest of the proteome (Fig. 3e, Supplementary Figure 6C, D, and Supplementary Data 5). These solubility effects differed substantially from those observed in 10 mM $MgCl_2$-treated samples, showing that ATP-induced solubility changes are predominantly not salt driven (Supplementary Figure 6E). However, the solubilizing properties of ATP were mimicked by GTP (Supplementary Figure 7, Supplementary Data 6), as well as by a non-hydrolysable analog of ATP (AMP-PNP) (Fig. 3c, Supplementary Figure 8, and Supplementary Data 7). These observations suggest that ATP hydrolysis is not the main factor for ATP-mediated solubilization and that previous findings based on fluorescently tagged nucleolus markers FBL and NPM1 cannot be generalized[37].

Furthermore, we used TPP to investigate how the addition of 10 mM ATP affected the thermal stability of solubilized proteins (Supplementary Data 8) and observed a significant trend for thermal destabilization compared to the vehicle condition (Fig. 3f). This suggests that the ATP-solubilized fractions of these proteins tended to have lower thermal stability than the soluble ones, which may be due to distinct interactions or post-translational modifications in the two populations.

**ATP depletion in cells decreases solubility of proteins.** To extend our analysis of the role of ATP on proteome solubility to cells, we again depleted ATP from Jurkat cells with 2-deoxyglucose and Antimycin-A (Fig. 4a). We identified 107 proteins with decreased solubility—14% of the insoluble proteome—in the ATP-depleted cells compared to vehicle condition

(Supplementary Data 9, Supplementary Figure 9A). As discussed previously, since ATP depletion is expected to affect several homeostatic processes, our analysis primarily sought to assess the changes in solubility in cells of proteins that gained solubility in the crude lysate system (Fig. 4a, Supplementary Figure 9B, C, Supplementary Data 9). In general, proteins solubilized at lower ATP concentrations in crude lysate had a stronger decrease in solubility upon cellular ATP depletion (Fig. 4b). For example, FBL, which is solubilized at higher ATP concentrations in crude lysate, showed no decrease in solubility in cells upon ATP depletion (Fig. 4c). In contrast, DDX50 and BANF1, two proteins that were solubilized with low concentrations of ATP in crude lysate, displayed decreased solubility following ATP depletion in cells (Fig. 2i). We thus conclude that the solubilizing effects of ATP are also relevant in cellular systems.

**ATP alters DNA-binding of BANF1.** Following the global analysis, we set out to understand the solubility changes observed in one of the highly ATP-solubilized proteins, Barrier to auto-integration factor (BANF1)—which is not annotated to be part of any membrane-less organelle. BANF1 is a non-specific DNA-binding protein, which transiently cross-bridges anaphase chromosomes to promote assembly of a single nucleus[38]. We hypothesized that ATP may affect its ability to bind to DNA and hence tested if purified BANF1 (Supplementary Figure 10) bound double stranded DNA in the presence and absence of ATP. We observed that the ability of BANF1 to interact with DNA was lower with increasing concentrations of ATP (Fig. 4d). The presence of 2.5 mM ATP reduced the amount of BANF1 bound to DNA by close to 50% (Fig. 4d). This suggests that ATP solubilizes BANF1 by preventing its binding to DNA, and thus represents an example of ATP regulating protein-DNA interactions.

**ATP lowers the solubility of a few proteins.** In crude lysates, a small number of proteins became less soluble with increasing ATP concentrations, with two prominent examples being IMPDH1 and NUCKS1 (Fig. 5a, Supplementary Data 5). In both cases, ATP hydrolysis is likely to play a role, since neither protein decreased in solubility following addition of the non-hydrolyzable ATP analog AMP-PNP (Supplementary Figure 8E). ATP was reported to act as an allosteric modulator of recombinant IMPDH1, driving the formation of a filamentous structure[39], in agreement with our observations on endogenous IMPDH1. NUCKS1 is predicted to be a highly disordered (99%), chromatin

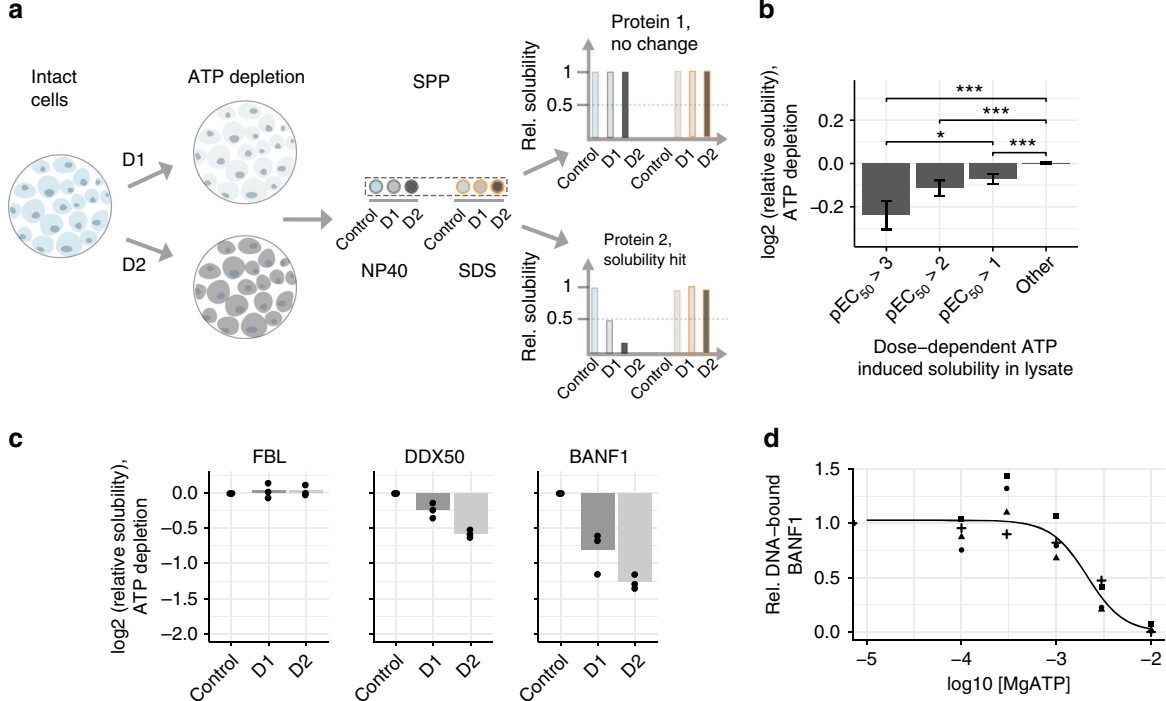

**Fig. 4** SPP of ATP-depleted cells recapitulates effects observed in crude lysate. **a** Experimental setup of SPP in ATP-depleted cells. Cells were depleted of ATP by inhibiting glycolysis and oxidative phosphorylation with two combinations of 2-deoxyglucose (2DG) and Antimycin-A (AA) (D1: 0.1 nM AA and 1 mM 2DG, D2: 1 nM AA and 10 mM 2DG). The untreated cells and the cells from the two ATP-depleted conditions were divided into two aliquots each, of which one was solubilized with NP40 while the other using 1% SDS. All samples were digested with trypsin, labeled with different TMT10 isotope tags and analyzed by LC-MS/MS. **b** Change in solubility calculated for proteins binned according to their propensity for solubilization with ATP in crude lysate ($pEC_{50}$ x-axis), following ATP depletion in cells (from three independt experiments). Error bars represent standard error of the mean. Significance levels obtained from a Wilcoxon signed-rank test were encoded as *$p < 0.05$, **$p < 0.01$, and ***$p < 0.001$. **c** Change in solubility of FBL, DDX50, and BANF1 following ATP depletion in cells, measured by calculating the $\log_2$ ratios of NP40-processed untreated and ATP-depleted conditions. **d** DNA-binding propensity of BANF1 in the presence of ATP. Recombinantly expressed and purified BANF1 was incubated with biotinylated double stranded DNA in the presence of increasing concentration of ATP. The DNA-bound fraction of BANF1 was pulled down using streptavidin beads and measured using quantitative mass spectrometry. Rel. DNA-bound fraction of BANF1 was calculated as the ratio protein bound to DNA in the presence of ATP compared to control sample without ATP. Data from four independent trials have been shown. Violin plots represent relative densities. For all box plots, the center line is the median, bounds of the boxes are the 75 and 25% percentiles (i.e., the IQR), and whiskers correspond to the highest or lowest respective value. If the lowest or highest value is an outlier (greater than 1.5*IQR from the bounds of the boxes) whiskers are exactly 1.5*IQR. Numbers above violin plots represent number of proteins. Source data are provided as a Source Data file for panels B-D

associated protein, which is necessary for DNA repair by homologous recombination[40]. Single nucleotide polymorphisms of NUCKS1 have been linked through genome wide association studies to Parkinson's, a protein aggregation-related disease[41]. The melting behavior of NUCKS1 in TPP experiments treated with 10 mM ATP followed a striking pattern, with a sharp increase of soluble protein abundance peaking at ~58 °C. At higher temperatures a temperature-dependent decrease of soluble protein was observed, with the melting curve matching that determined in the vehicle condition (Fig. 5b). This may suggest that the ATP-related metabolism induces NUCKS1 to transition into two-phases, which is observed as loss in solubility of the protein. Upon heating, the NUCKS1 assembly transitions into a single phase at 58 °C, indicative of its critical transition temperature[42]. In contrast, IMPDH1 exhibits a more gradual increase in solubility at lower temperatures, and shows a substantially higher thermal stability, indicating that the ATP-binding[39] leads to a more stable conformation (Fig. 5b). Finally, we asked ourselves if proteins that have insoluble subpopulations under physiological conditions (without addition of ATP) would in general show some increase in solubility upon heating. In order to answer that question, we looked specifically at maximal fold changes across the heating temperatures—calculated using 37 °C as

reference—observed for melting curves of proteins that have an insoluble subpopulation in the crude lysate (ratio of SDS/NP40 > 1.5). Indeed, for this group of proteins we saw a significant increase in the amount of protein that became soluble upon heating (Fig. 5c), indicating temperature-dependent phase transitions on a proteome-level. These observations suggest that the combination of thermal stability and solubility could potentially be used to characterize phase transitions on a system-wide scale.

## Discussion

Our findings map the proteome-wide regulatory landscape of ATP by measuring thermal stability and solubility of proteins from mechanically disrupted cells. We quantified ATP-protein interactions, which encompass its classical roles as a substrate, allosteric modulator, biosynthetic precursor, and competitive inhibitor along with elucidating its non-classical roles on the influence protein assemblies. Together, our TPP, 2D-TPP, and SPP data revealed concentration-dependent roles of ATP. At low concentrations (below 500 μM), ATP mainly acted as a substrate for nucleotide-binding proteins. At intermediate concentrations (between 1–2 mM), broad effects of ATP on the stability of protein complexes were observed. At higher concentrations

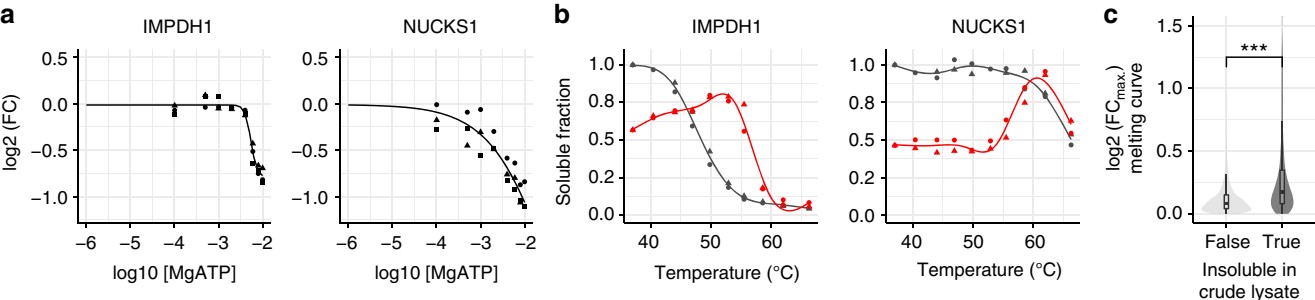

**Fig. 5** SPP identifies proteins decreasing in solubility upon ATP addition. **a** Solubility profile for IMPDH1 and NUCKS1, fold change (FC) in y-axis represents the ratio of NP40-processed ATP treated and vehicle-treated conditions from three independent experiments. **b** Melting curves for IMPDH1 and NUCKS1 from two independent TPP experiments in crude lysates treated with vehicle (black) or 10 mM ATP (red) and corrected for solubility changes at 10 mM ATP using SPP data. **c** Maximal log2 fold changes observed for melting curves of insoluble versus soluble proteins in untreated crude lysate. Significance levels obtained from a Wilcoxon signed-rank test were encoded as *$p < 0.05$, **$p < 0.01$, and ***$p < 0.001$. Violin plots represent relative densities. For all box plots, the center line is the median, bounds of the boxes are the 75 and 25% percentiles (i.e., the IQR), and whiskers correspond to the highest or lowest respective value. If the lowest or highest value is an outlier (greater than 1.5*IQR from the bounds of the boxes) whiskers are exactly 1.5*IQR. Numbers above violin plots represent number of proteins. Source data are provided as a Source Data file for all panels

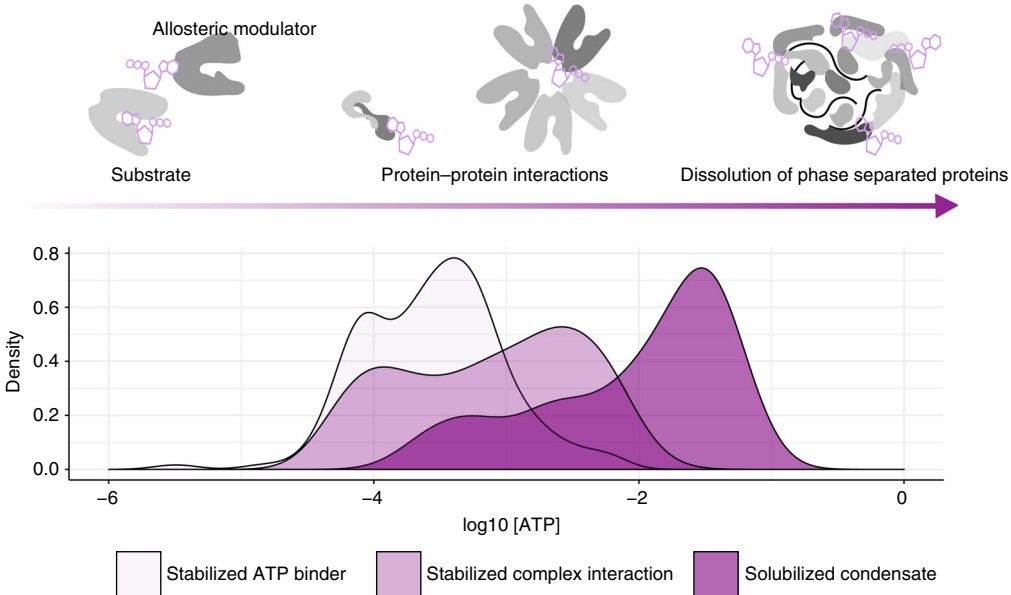

**Fig. 6** Concentration-dependent proteome-wide effect of ATP. Density plots of pEC$_{50}$s of ATP-stabilized ATP-binding proteins, complexes, as well as of proteins solubilized by ATP as measured by 2D-TPP and SPP technologies. Despite a clear difference in the distributions, there is a substantial overlap, showing that changes in ATP concentrations will have simultaneous effects on the stability of ATP-binding proteins and complexes as well as on the solubility of disordered proteins

(greater than 2 mM) ATP has solubilizing effects, primarily on disordered proteins with positive charge that bind nucleic acids such as DNA and RNA (Fig. 6). While high affinity interactions of ATP seem to be selective (except for a small subset of ATPases, which exhibit promiscuity between ATP and GTP), the low affinity solubility effect of ATP is mirrored with equivalent amounts of GTP.

The use of lysates obtained from mechanically disrupted cells has enabled us to assess the impact of ATP on both soluble and insoluble proteomes simultaneously. Our SPP analysis revealed that ATP solubilizes a substantial fraction of the insoluble proteome that is enriched for components of membrane-less organelles. These organelles are macromolecular condensates formed by weak interactions of proteins and nucleic acids by a phenomenon called phase separation[43]. Previous studies suggested that active processes utilizing ATP drive the regulation of such assemblies[6,8,44]. However, our data strongly suggest that ATP regulates phase separation not only by modulating enzyme activities, but also by directly promoting dissolution of such condensates. Recent reports suggest that long-range electrostatic and short-range cation-π or π–π interactions in the protein sequence may be critical for phase separation[45,46]. The aromatic adenine ring and highly negatively charged phosphoanhydride bond of ATP may disrupt π–π and electrostatic interactions in positively charged proteins to increase their solubility. Furthermore, the ATP-solubilized subpopulation of proteins exhibits lower thermal stability compared to their soluble counterparts. This suggests that, although mechanisms independent of ATP hydrolysis can solubilize proteins, appropriate post-translational modifications may be required for their stability in the soluble state.

ATP seems to influence nucleic acid binding properties of proteins. BANF1 binds DNA nonspecifically and is a key protein involved in nuclear organization. Phosphorylation of BANF1 has been reported to regulate its DNA-binding properties[47]. However, we observed ATP-induced solubilization of BANF1 in crude lysate experiments, and drastic reduction in its solubility upon

ATP depletion in cells. Furthermore, reduction in DNA-bound fraction of BANF1 (recombinantly purified) with increasing dose of ATP, demonstrates the role of ATP in regulating non-specific nucleic acid-protein interactions. These observations suggest that solubility of BANF1 can be modulated independently of phosphorylation by altering ATP levels. Previous studies show that ATP depletion causes chromatin compaction[48], we speculate that BANF1 may interact with chromatin and compact DNA at low ATP levels.

Taken together, the ATP-dependency exhibited on a proteome-level suggests that metabolic fluctuations, such as nutrient deprivation, that affect ATP concentrations in cells, may be sensed as the altered solubility status of nucleic acid binding proteins. Previous studies using single particle tracking showed that solubility of cytoplasmic proteins of both prokaryotes and eukaryotes are reduced upon cellular stress[34,49]. Our data put forth the hypothesis that ATP levels in cells may have a direct impact on the dynamics of protein-nucleic acid interactions, which may explain the consequential drops in rates of transcription and translation in cells. Our data provide a unique map of the subproteomes affected in their solubility and thermal stability by ATP and will facilitate the discovery of ATP-binding proteins, components of membrane-less organelles, and regulators of phase separation. The SPP approach described herein can be applied in future studies to assess the effects of other molecules and cellular perturbations on protein solubility in different disease-relevant cell types, to study regulation of aggregation-susceptible proteins and organelles.

## Methods

**Materials**. All chemicals were purchased from Sigma–Aldrich unless otherwise specified.

**Cell culture**. Jurkat E6.1 cells (obtained from ATCC) were cultured in RPMI media (GIBCO) containing 10% heat inactivated fetal calf serum to a maximum density of $2 \times 10^6$ cells/ml.

**Preparation of crude cell extracts**. Jurkat cells were harvested and washed with phosphate-buffered saline (PBS, 2.67 mM KCl, 1.5 mM $KH_2PO_4$, 137 mM NaCl, and 8.1 mM $NaH_2PO_4$, pH 7.4). The cell pellet was resuspended in lysis buffer (PBS containing protease inhibitors and 1.5 mM $MgCl_2$) equal to 10 times the volume of the cell pellet. The cell suspension was lysed by mechanical disruption using a Dounce homogenizer (20 strokes). Protein concentration of the resulting crude extract was measured using Bradford assay. The crude extract was diluted to 3.5 mg/ml for all subsequent steps.

**Preparation of gel-filtered lysate**. The crude extract of Jurkat cells was treated with benzonase (25 U per ml—final concentration) for 60 min at 4 °C on a shaking platform (650 rpm). Subsequently, the extract was centrifuged at $100,000 \times g$ for 20 min at 4 °C. The supernatant cleared of cell debris was desalted using PD-10 column (GE, healthcare) using manufacturer's instructions to deplete endogenous metabolites. In short, the PD-10 columns were equilibrated with 5 column volumes of lysis buffer before the application of 1.5 ml cleared lysate. The sample was eluted by centrifugation (1000 × g for 1 min at 4 °C). Further, the protein concentration of the gel-filtered lysate was measured using Bradford assay and lysate was diluted to 2 mg/ml for all subsequent steps.

**ATP measurements in cell extracts**. ATP levels in crude and gel-filtered lysate was estimated using Cell-Titer Glo assay (Promega) using manufacturer's protocol. In short, equal parts of cell extract (or ATP standards) and cell-titer glo reagent were mixed and incubated on a shaking platform (500 rpm) for 30 min and the luminescence readout was measured. In order to measure the hydrolysis rates of ATP in the experimental setup, lysis buffer (control), crude, and gel-filtered lysates were incubated with 2 mM Na-ATP for 10 min at room temperature and an endpoint measurement of relative ATP levels in the extracts compared to control was measured.

**Thermal proteome profiling in cell extracts**. Na-ATP, Mg-ATP, and Na-GTP stocks (300 mM) were prepared in PBS containing 1.5 mM $MgCl_2$ and the pH was adjusted to 7 using NaOH.

For TPP-temperature range (TPP-TR) experiments[11], two parts of the lysates (both crude and gel filtered), one treated with metabolite (2 mM Na-ATP or 10 mM Mg-ATP or 0.5 mM GTP—final concentration) and other treated with the vehicle (PBS with 1.5 mM $MgCl_2$), were used. After a 10-min incubation at room temperature, the metabolite- and vehicle-treated lysates were split into 10 aliquots and heated to different temperatures (37.0, 40.4, 44.0, 46.9, 49.8, 52.9, 55.5, 58.6, 62.0, and 66.3 °C) for 3 min. Following heat treatment, the crude lysate samples were treated with NP40 and benzonase (final concentration—0.8% and 25 U per ml, respectively) for 60 min at 4 °C on a shaking platform (650 rpm) to extract membrane-bound and DNA-bound proteins. The protein aggregates for both the lysate types was removed by ultracentrifugation (100,000 × g, 4 °C, 20 min) and the supernatant was collected. TPP-TR, lysate experiments were conducted as two biological replicates.

2D-TPP experiments[14] were performed using crude lysate. Briefly, crude lysate was treated with five concentrations (including vehicle control) of Na-ATP (0, 0.005, 0.05, 0.5, and 2 mM) or Na-GTP (0, 0.001, 0.01, 0.1, and 0.5 mM) for 10 min at room temperature. Samples from each metabolite concentration were split in 12 aliquots and were heated to different temperatures (42, 44.1, 46.2, 48.1, 50.4, 51.9, 54.0, 56.1, 58.2, 60.1, 62.4, and 63.9 °C) for 3 min. Protein aggregates were removed by ultracentrifugation (100,000 × g, 4 °C, 20 min) after the heat-treated lysate was incubated with NP40 and benzonase ((final concentration—0.8% and 1 U per ml, respectively) at 4 °C for 60 min on a shaking platform (650 rpm). The soluble supernatant was collected. Three independent experiments were performed for robust data analysis.

**Solubility proteome profiling in cell extracts**. The crude lysate was split into 10 aliquots. Two of ten were used as vehicle-treated controls. The remaining eight aliquots were treated with increasing concentrations (0.1, 0.5, 1, 2, 4, 5, 8, and 10 mM) of a small molecule (Mg-ATP or Mg-AMP-PNP or Mg-GTP) for 10 min at room temperature, followed by 3 min at 37 °C. To one vehicle-treated sample and the eight small molecule-treated sample, NP40 (to a final concentration of 0.8%) was added, while to the other vehicle-treated aliquot SDS (final concentration of 1%) was added. Subsequently all 10 aliquots were treated with benzonase (25 U per ml at 4 °C for 60 min). The insoluble proteins were removed by ultracentrifugation (100,000 × g at 4 °C for 20 min) and the supernatant was collected. All SPP experiments were conducted as three independent experiments.

**Compound treatment and cellular ATP measurements**. Glycolysis inhibitor 2-deoxyglucose (2DG) was dissolved in water and oxidative phosphorylation (complex III) inhibitor Antimycin-A was dissolved in 95% ethanol and 50X stock of the treatment concentrations (see below) were prepared. A combination of both inhibitors was used to reduce ATP levels in cells. Jurkat cells were washed with PBS and resuspended in RPMI media (US biologicals) without glucose and FBS at the density of $2 \times 10^6$ cells per ml and split into three portions. To the first portion, glucose (10 mM—final concentration) was added along with vehicle of the inhibitors. The second portion was treated with 0.1 nM Antimycin-A and 1 mM 2DG (called as D1 condition) and the third portion was treated with 1 nM Antimycin-A and 10 mM 2DG (D2 condition). The three plates were incubated at 37 °C with 5% $CO_2$ for 60 min. At the end of 60 min, 200,000 cells for each condition (100 µl cells) was incubated with 100 µl of Cell-Titer Glo reagent on a shaking platform for 30 min and the luminescence was measured to calculate the relative levels of ATP in D1 and D2 conditions with control cells as the reference.

**TPP-TR experiments in cells**. At the end of 60 min with the above mentioned inhibitors, the cells were pelleted by a short centrifugation (1000 × g for 3 min). The cell pellet resuspended in media (same as treatment) to bring the cell density to $2 \times 10^7$ cells per ml. Cells (100 µl) were split into 12 aliquots, of which two were used for SPP sample preparation (see below). The remaining 10 aliquots of cells were spun at 1000 × g, 3 min and 80 µl of media was removed. Each aliquot was heated to different temperatures (37.0, 40.4, 44.0, 46.9, 49.8, 52.9, 55.5, 58.6, 62.0, and 66.3 °C) for 3 min. Following heat treatment, the cells were lysed with 50 µl lysis buffer (PBS containing protease inhibitors, 1.12% NP40, 2.1 mM $MgCl_2$, phosphatase inhibitor, and 35U per ml Benzonase) and incubated at 4 °C for 60 min on a shaking platform (650 rpm). The protein aggregates and cell debris were removed by ultracentrifugation (100,000 × g, 20 min at 4 °C) and the supernatant was collected.

**SPP in cells**. The two cell aliquots (collected as described above), were spun at 1000 × g, 3 min and 80 µl of the media was removed. One aliquot was lysed with a mild detergent (PBS containing protease inhibitors, 1.12% NP40, 2.1 mM $MgCl_2$, phosphatase inhibitor, and 35 U per ml Benzonase), while the other was lysed with a strong detergent (PBS containing protease inhibitors, 1.4% SDS, 2.1 mM $MgCl_2$, phosphatase inhibitor, and 35 U per ml Benzonase). The insoluble proteins were removed by ultracentrifugation at 100,000 × g, 20 min at 4 °C.

**Protein digestion and peptide labelling**. From 20 to 25 µg of proteins (based on two lowest temperatures in case TPP experiments, and NP40-processed vehicle-treated samples in case of SPP experiments) from the supernatant were reduced, alkylated, and digested with Trypsin and LysC using modified SP3 protocol[50]. Briefly, proteins samples were bound to acidified paramagnetic beads (0.5 µg Sera-Mag-beads A and B in 10 µl of 15% formic acid and 30 µl ethanol). Following

an incubation of 15 min, the protein-bound beads were washed four times with 70% ethanol. On bead reduction, alkylation, and digestion of the proteins were performed overnight using 0.25 µg Trypsin, 0.25 µg LysC, 1.7 mM TCEP, and 5 mM chloroacetamide in 100 mM HEPES, pH 8. Subsequently, the peptides were eluted from the beads and lyophilized. The dried peptides were dissolved with 10 µl water and 10 µl of TMT labels (8 µg per µl) was added and incubated for 60 min. The labelling reactions were quenched with 5 µl hydroxylamine (2.5%) solution and pooled together and vacuum dried for LC-MS-MS analysis.

For TPP-temperature range experiments, sample from 10 different temperatures were labelled with 10 different TMT tags and pooled together as a single TMT experiment. In a 2D-TPP experiment, the concentration range of two neighboring temperatures were labelled and pooled as a single TMT experiment. In case of SPP experiment, nine NP40-processed (one vehicle and eight small molecule treated) samples and one SDS-processed (vehicle) sample were labelled and pooled as a single TMT experiment.

Dried peptide samples were reconstituted in 1.25% ammonia in water and fractionated on an Ultimate3000 (Dionex) by reversed-phase chromatography at pH 12 (Buffer A: 20 mM Ammonium formate pH 10, Buffer B: Acetonitrile) on X-bridge column (2.1 × 10 mm, C18, 3.5 µm, Waters, Milford, MA), and 24 fractions were collected and vacuum dried.

### LC-MS/MS measurements
Peptides were resuspended in 0.05% formic acid and analyzed on Q Exactive Orbitrap or Orbitrap Fusion Lumos mass spectrometers (Thermo Fischer Scientific) as described in[15,35]. In short, peptides were separated using an Ultimate 3000 nano RSLC system (Dionex) equipped with a trapping cartridge (Precolumn C18 PepMap100, 5 mm, 300 µm i.d., 5 µm, 100 Å) and an analytical column (Acclaim PepMap 100. 75 × 50 cm C18, 3 mm, 100 Å). A nanospray-Flex ion source was used. 0.1% formic acid prepared with LC-MS grade water (Fischer Scientific) was used as Solvent A and 0.1% formic acid in LC-MS grade acetonitrile (Fischer Scientific) served as Solvent B for chromatographic preparation. The peptides were loaded onto the trap column at 30 ml per min of solvent A and eluted using a gradient from 2 to 40% Solvent B over 2 h at 0.3 ml per min.

The Q Exactive Plus was operated in positive ion mode. Full scan MS spectra with a mass range of 375–1200 m/z were acquired in profile mode using a resolution of 70,000 (maximum fill time of 250 ms or a maximum of 3e6 ions (automatic gain control, AGC)). Fragmentation was triggered for the top 10 peaks with charge 2–4 on the MS scan (data-dependent acquisition) with a 30 s dynamic exclusion window. Precursors were isolated using a mass window of 0.7 mass by charge (m/z), fragmented using 33 NCE (normalized collision energy) and MS/MS spectra were acquired in profile mode with a resolution of 35,000 (at m/z 200, and ion target setting was set to 2e5 ions to avoid coalescence)[19].

The Orbitrap Fusion Lumos was operated in positive ion mode with a spray voltage of 2.4 kV and capillary temperature of 275 °C. Full scan MS spectra with a mass range of 375–1500 m/z were acquired in profile mode using a resolution of 120,000 (maximum fill time of 50 ms or a maximum of 4e5 ions (AGC) and a RF lens setting was 30%. Fragmentation was triggered for 3 s cycle time for peptide like features with charge states of 2–7 on the MS scan (data-dependent acquisition). Precursors were isolated using the quadrupole with a window of 0.7 m/z and fragmented with a normalized collision energy of 38. Fragment mass spectra were acquired in profile mode and a resolution of 30,000 in profile mode. Maximum fill time was set to 64 ms or an AGC target of 1e5 ions). The dynamic exclusion was 45 s.

### Recombinant expression and purification of BANF1
Codon-optimized gene encoding BANF1 for E.coli (sequence provided in Supplementary Data 10) inserted between NcoI and HindIII sites of pET28a vector was purchased from Thermoscientific. BANF1 was expressed in E.coli-BL21 (DE3) strain as a fusion protein with His tag-Thioredoxin tag-TEV site-BANF1. The cells were grown at 24 °C for 24 h in terrific broth containing 2% lactose. The cells were harvested and lysed under native conditions (50 mM Tris-Cl, 250 mM NaCl, 20 mM Imidazole, pH 8 containing protease inhibitors and nuclease) by sonication. The soluble protein fraction obtained after high speed centrifugation (100,000 × g, 30 min, 4 °C) of lysate was subjected to affinity purification using Ni-NTA column. In short, the fusion protein from the soluble protein fraction was passed through $Ni^{2+}$-affinity column for binding, following a thorough wash step, the bound fraction was eluted using 200 mM imidazole in 50 mM Tris-Cl, 250 mM NaCl, pH 8 buffer (Supplementary Figure 10a, b). The eluted fraction enriched in the fusion protein was dialyzed using 50 mM Tris-Cl, 250 mM NaCl, and 10 mM Imidazole pH 8 overnight in the presence of TEV protease. The His-Thioredoxin-tag and uncut fusion protein were separated from the protein was interest using Ni-NTA affinity chromatography. Further, BANF1 was purified by gel filtration chromatography using Superdex-75 column (GE Healthcare), where it eluted as dimer. The protein fractions containing BANF1 were pooled and quantified using BCA assay and stored in 50 mM Tris-Cl, 150 mM NaCl, 5% Glycerol pH 7.5. The purity of protein sample was assessed by measuring the intact mass of BANF1 on Q-TOF mass spectrometer (Supplementary Figure 10C).

### BANF1-DNA-binding assay
DNA-BANF1 pull down assays were performed using Biotinylated oligos described previously used[51] (FW-5'-Biotin-GTGTGGAAAATCTCTAGCAGTAAAAAAAAAA-3' and RV-5'-TTTTTTTTTTTACTGCTAGAGATTTTCCACAC-3'). Equimolar concentrations of FW and RV oligo were mixed and annealed (95 °C—5 min, cool down 4 °C at 0.1 °C per sec). Double stranded DNA oligo was incubated with purified BANF1 at 1:1 ratio in the presence of 0, 0.1, 0.3, 1, 3, and 10 mM Mg-ATP in Tris-buffered saline, 10 mM Tris-Cl, 150 mM NaCl, 3 mM KCl pH 7.5 (TBS) at 25 °C for 30 min. DNA-protein complex was pulled down using my-One Streptavidin beads (Thermoscientific) using manufacturer's protocol. Only the purified protein incubated with the streptavidin beads was used for correcting non-specific binding of protein to beads. The beads were washed with TBS containing 0.2% NP40 buffer with or without ATP and the bound protein was eluted in 0.1% SDS containing TBS followed by heating the beads at 95 °C for 5 min. The eluted protein fraction (non-specific binding controls, DNA-protein positive controls, DNA-Protein-ATP samples in triplicate) was reduced, alkylated, and digested using 0.02 µg Trypsin, 0.02 µg LysC, 1.7 mM TCEP, and 5 mM chloroacetamide in 100 mM HEPES, pH 8 at 37 °C for 4 h after which the peptides were lyophilized. The dried peptides were dissolved with 10 µl water and 5 µl of TMT labels (8 µg/µl) was added and incubated for 60 min. The labelling reactions were quenched with 5 µl hydroxylamine (2.5%) solution and pooled together and vacuum dried for LC-MS/MS analysis.

### Peptide and protein identification
In order to identify peptides and proteins, raw MS-data were processed using a combination of Isobarquant[13,15,20] with MASCOT 2.4 (Matrix Science). Using the following parameters, the data were searched against a human database from Uniprot downloaded on 14 May 2016 or Swissprot downloaded on 10 January 2018 combined with the decoy version of the database: trypsin, missed cleavage 3, peptide tolerance 10 ppm, 0.02 Da for MS-MS tolerance, fixed modifications included carbamidomethyl on cysteines and TMT 10plex on lysine, variable modifications included acetylation of protein N-terminus, methionine oxidation and TMT10plex on peptide N-termini. A minimum of two unique peptides with a peptide length of at least seven amino acids and a Mascot score of greater than 15 as well as an FDR below 0.01 were required for each protein. The protein FDR was set at 0.01.

### TPP-temperature-range data
Data from TPP-TR experiments were analyzed using the TPP package[20], and melting points of the proteins were determined

### 2D-TPP data preprocessing
2D-TPP datasets were preprocessed and normalized as previously described[14]. This included computation of fold changes to respective vehicle conditions per temperature and median normalization of these values within one TMT label.

### FDR-controlled detection of ligand-protein interactions
In the 2D-TPP strategy, aliquots of cells or lysates are treated with different concentrations of a drug as well as a vehicle condition. Therefore, it is reasonable to expect that if a protein's stability is affected by the drug treatment, the response will result in a dose-dependent (de-) stabilization. In particular, we expect a dose-dependent effect on stability for two consecutive temperatures[14]. To capture this, parametric dose-response curves[52] were fitted to relative stability values of each protein $i$ in sliding windows of adjacent temperatures $j$ per replicate $r$ (Supplementary Figure 2A) with:

$$\mu_{i,j,r}^{H_1}(x) = c + \frac{d - c}{1 + \exp(b(x - e))} \quad (1)$$

Here, $x$ corresponds to the $\log_{10}$ transformed drug dose, $b$ is the slope, $c$ the intercept, $d$ the maximal value, and $e$ the positive $\log_{10}$ transformed effective concentration ($pEC_{50}$). The parameters $b$, $c$, $d$, and $e$ are estimated from the data. In parallel, the same data points were fitted by a simple intercept model expected under the null hypothesis, $H_0$, of no dose-dependent stabilization:

$$\mu_{i,j,r}^{H_0} = c \quad (2)$$

Residual sum of squares (RSS) of both fits were used to compute a $F$-statistic per protein $i$ and temperature window $j$, similar to a previous method for microarray time course experiments[53]:

$$F_{i,j} = \text{median}_r \left( \frac{RSS_r^{H_0} - RSS_r^{H_1}}{RSS_r^{H_1}} \right) \quad (3)$$

To obtain a per protein score $F_i^{\text{comb.}}$ combining scores obtained from all temperature windows $j$, we computed:

$$F_i^{\text{comb.}} = \sum_{j \in J_{\text{thres.}}} \log_e(F_{i,j}) \quad (4)$$

where

$$J_{\text{thres.}} = \left\{ j \in J | \varphi_j^{\max} > h \text{ or } \varphi_j^{\max} < \frac{1}{h} \right\}$$

where $\varphi_j^{\max}$ is the maximum relative fold change observed in a window $j$, and $h$ is a predefined threshold (set at 1.5 throughout this work). The logarithm transformation has the effect that $F_i^{\text{comb.}}$ corresponds to a geometric mean, which gives less leverage to outlier values of $F_{i,j}$.

In order to estimate the false discovery rate (FDR)[54] for $F_i^{\text{comb.}}$, we permuted the dataset 100 times as follows: Relative stability values, except the reference values (vehicle treatment), at the same temperature and within groups of peptides used for quantification (qupm $\in \{2, 3, 4, 5 \text{ or more}\}$) were randomly permuted across proteins and treatment concentrations (Supplementary Figure 2B). $F_i^{\text{comb.}}$ scores as described above were computed for each of the 100 permutations and ranked jointly with the value from the original data. Based on these individual rankings an average FDR was computed for each possible threshold $\theta$ across all $B$ permutations with:

$$\text{FDR}_\theta = \frac{1}{B} \sum_{b=1}^{B} \frac{2v_b}{r + v_b}. \qquad (5)$$

Here $r$ corresponds to the number of hits obtained with the original dataset and $v_b$ to the number of hits with permutated dataset $b$.

As a measure of ligand-protein affinity, we assigned each protein the median $\text{pEC}_{50}$ retrieved at the lowest temperature where a minimal fold change of $h$ in case of stabilization or $\frac{1}{h}$ in case of destabilization was observed in at least two replicates.

**SPP data preprocessing and normalization**. Summed ion area intensities obtained from SPP experiments were subjected to variance-stabilizing normalization (vsn)[55] across replicates and conditions. Retrieved values were back transformed from $\log_2$ space and fold changes with regard to vehicle conditions of individual replicates were computed.

**FDR-controlled detection of protein solubility changes**. A similar approach as for the analysis of 2D-TPP experiments was applied to SPP datasets. For each protein $i$ parametric dose-response models $\mu_i^{H_1}(x)$ were fitted (Eq. 1) across all replicates and compared with intercept models $\mu_i^{H_0}(x)$ (Eq. 2) and a $F$-statistic was deduced with:

$$F_i = \frac{\text{RSS}^{H_0} - \text{RSS}^{H_1}}{\text{RSS}^{H_1}}. \qquad (6)$$

In order to estimate FDR, the true dataset was permuted 100 times as described for the 2D-TPP analysis and the same analysis to obtain F-statistics as described above was performed for each of the permuted datasets. $F$-statistics retrieved from permuted dataset, ranked jointly and an average FDR, was computed. We applied a cutoff of 1% FDR and requested a minimal median fold change of $h$ in case of solubilization or $\frac{1}{h}$ in case of de-solubilization ($h = 1.5$).

**Gene ontology annotation**. To visually represent different groups of proteins that were found to be stabilized by either ATP or GTP, proteins were annotated to GO molecular function (MF) terms in a non-overlapping manner. GO terms ATP-binding (GO:0005524), protein kinase activity (GO:0004672), and ATPase activity (GO:0016887) were aggregated to ATP binding. Likewise the terms GTP binding (GO:0005525) and GTPase activity (GO:0003924) were aggregated to GTP binding, NADPH binding (GO:0070402), NADP + binding (GO:0070401), NAD binding (GO:0051287), NAD + binding (GO:0070403), FAD binding (GO:0071949), flavin adenine dinucleotide binding (GO:0050660) to NAD/FAD binding, RNA binding (GO:0003723) and DNA binding (GO:0003677) to RNA/DNA binding, and kinase binding (GO:0019900), ATPase binding (GO:0051117), GTPase binding (GO:0051020) and if they were part of a protein complex (annotation described below) to part of complex & reg. subunits. Proteins that did not match any on these annotations were categorized as other.

**Gene ontology enrichment**. The enrichment analysis of GO cellular compartment (CC) terms for proteins exhibiting ATP/AMP-PNP/GTP dose-dependent solubilization was performed using clusterProfiler (R Bioconductor)[56] by taking all identified proteins in the experiment as the background. Standard settings were used ($p$-value cutoff: 0.05, Benjamini-Hochberg procedure for multiple testing adjustment and $q$-value cutoff of 0.2).

**Sequence motif enrichment**. Full sequences of all proteins stabilized by ATP and GTP were retrieved from UniProt using the R UniProt.ws package. Sequences containing the classical walker-A motif GXXXXGK were extracted along with 20 amino acid residues flanking the motif. Using ggseqlogo[57], a R package, motif analysis was performed including six amino acids upstream and eight amino acid downstream of the core motif. The same analysis was performed for only those proteins found to be stabilized with high potency ($\text{pEC}_{50} > (\text{mean}(\text{pEC}_{50}) + \text{SD}(\text{pEC}_{50}))$). Proteins were assigned to groups according to the presence or absence of a basic residue (lysine (K), histidine (H), or arginine (R)) at the $-5$ position in the upstream flanking region of their walker. Distributions of $\text{pEC}_{50}$ values obtained from 2D-TPP experiments with GTP were then tested from mean group differences using Wilcoxon-rank test.

**Protein domain annotation**. Protein domain enrichment analysis was carried out using InterPro database[58] on DAVID platform (version 6.8)[59] for all proteins found to be stabilized in 2D-TPP datasets of ATP and GTP. For visualization purposes, the $\text{pEC}_{50}$ distribution of proteins in following categories have been shown: P-loop—IPR027417, ATPase—includes annotations from IPR018631, IPR003593, and IPR001757, Kinase—all terms within IPR011009, OB-fold contains both IPR0012340 and IPR012940, NAD/FAD binding includes IPR036291 and IPR002938, and small GTP binding consists of IPR005225, IPR001019, and IPR016433.

**Protein complex analysis**. Proteins that belong to a complex were annotated as described in ref. [60] and were filtered to contain at least three quantified subunits. The Euclidean distance of every protein-pair in a complex was calculated from the melting profiles generated from the vehicle-treated crude lysate TPP-temperature range dataset as described in ref. [33]. Based on the comparison between average Euclidean distance of protein complexes and all other proteins in the dataset, a Euclidean distance of 0.02 (corresponding to $p < 0.05$ obtained by comparison to sampling of random pairwise protein comparisons) between two proteins was chosen as cutoff to define protein pairs that exist in proximity, which results in co-melting.

**Disordered proteins and protein isoelectric point mapping**. Using D2P2 database[61], which integrates protein disorder prediction from multiple algorithms with structure region prediction to robustly map the predicted disordered regions in protein, the information on the relative disorder rank of proteins (based on the number of amino acid residues in the predicted disordered region) was collated.

The average isoelectric points of proteins were mapped based on proteome-pI database[62].

**Reporting summary**. Further information on experimental design is available in the Nature Research Reporting Summary linked to this article.

**Code availability**. An implementation of the above FDR-controlled procedures can bet found at https://git.embl.de/kurzawa/sw2dTPP.

## Data Availability
All raw mass spectrometry data have been deposited on Proteomics Identifications (PRIDE)[63] database. Accession numbers are: PXD012423 (contains all TPP experiments), PXD012356 (contains all SPP experiments), and PXD012745 (contains effect of Mg on proteome solubility data). Supplementary Data 1–9 provide normalized data used for analysis and interpretation. The source data underlying Figs. 1B–G, 2B–D, 3B–F, 4B–D, 5A–C, 6 and Supplementary Figs. 1B, C, 3A–J, 4A–E, 5B, C, 6A–E, 7A–D, 8A–E, 9A–D, 10B, C are provided as a Source Data file. A reporting summary for this Article is available as a Supplementary Information file. All other data supporting the findings of this study are available from the corresponding authors on reasonable request.

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

## Acknowledgements
We thank André Mateus and Life Science Editors for insightful discussions and proof-reading of the manuscript, Natalie Romanov for supplying and helping with protein complex and disorder annotations; the Proteomics Core Facility at the European Molecular Biology Laboratory (EMBL) for expert help; K. Remans, J. Scheurich, and J. Flock from Protein Expression and Purification Core Facility at EMBL for help in recombinant expression and purification of BANF1, Friedrich Reinhard and Christina Rau at Cellzome-GSK for help with the crude lysate protocol and J. Stuhlfauth and N. Garcia-Altrieth at Cellzome-GSK for cell culture; M. Jundt, K. Kammerer, M. Klös-Hudak, M. Paulmann and T. Rudi at Cellzome-GSK for expert technical assistance.

## Author contributions
S.S., N.K., M.B. and M.M.S. conceptualized the study, S.S., T.W., M.B., and M.M.S. designed the experiments, S.S., T.W., I.G. and D.H. performed the experiments, N.K., W.H., M.B. and M.M.S. designed and implemented data analysis strategies, S.S., N.K., M.B. and M.M.S. analyzed the data, S.S., N.K., M.B. and M.M.S. wrote the manuscript.

## Additional information

**Competing interests:** S.S., T.W., I.T. and M.B. are employees and/or shareholders of Cellzome GmbH and GlaxoSmithKline. The remaining authors declare no competing interests.

