## [Peer Review File · Nature Communications]

REVIEWERS' COMMENTS:

Reviewer #1 (Remarks to the Author):

In recent years, the non-canonical role of metabolites is gaining increasing importance in cell biology. Building on recent literature, Sridharan et al. present a compelling study for the distinct roles of the molecule ATP on a proteome level by designing a novel method to complement biochemical and proteomic techniques. This study clearly suggests that in addition to the well-known function of ATP as the energy currency of the cell, it can exhibit additional regulatory roles by modulating the solubility of proteins. It is the first to show a general role of ATP in keeping proteins soluble, and the types of proteins that are solubilised by ATP. I strongly recommend the publication of this manuscript after minor improvements.

- 1) In their control experiments of untreated samples authors should use equivalent salt to control for the salt like property of ATP. Or clearly describe what exactly are the conditions in “vehicle treated” controls?
- 2) The authors always represent the concentration of ATP in the log scale in all the graphs and figures. They should represent the real concentration of ATP to make the figures more coherent with the text and easier to understand.
- 3) Depending on amount of effort needed, it will be beneficial to be consistent with the use of ATP-Mg and ATP-Na. Or the authors should include a statement on why ATP-Na was used for thermal proteome profiling (TPP) and ATP-Mg for solubility proteome profiling (SPP).
- 4) In the text the explanation for choosing the candidates in Fig 1 B-C comes after Fig 1E. Authors should try to keep the text and figures chronological.
- 5) In the schematic figures for 2D-TPP and SPP (Fig 1A and 2A) it is not quite clear what the different colors and intensities are representing?
- 6) The authors should pay attention to the numbers that are mentioned in the text are consistent with ones mentioned in the graphs (line 61: 315 ATP binders/ Fig 1D – 313 ATP binders).
- 7) In a recent paper (J. Kang et al., Biochemical and Biophysical Research Communications 504, 2018), it was pointed out that low concentrations of ATP in micromolar range induce phase separation of the ALS-linked RNA binding protein FUS while millimolar concentrations disrupt phase separation. Will be interesting to observe if this is some general phenomena across several proteins, that might explain a positive feedback of aggregation driven by a decrease in ATP concentration.

Reviewer #2 (Remarks to the Author):

This manuscript uses 2D thermal proteome profiling (TPP) to map NTP-interactions on a proteome-wide scale and solubility proteome profiling (SPP) to examine the role of NTPs in protein solubility. With TPP experiments, the authors were able to detect known ATP and GTP-binding proteins as well as identify proteins that are not currently classified as ATP or GTP-binding proteins. The authors also demonstrated that results found in crude lysates could be recapitulated in intact cells. The results also showed that subunits of proteins that do not directly bind ATP or GTP are stabilized when they are in close proximity to subunits that are bound and stabilized by ATP or GTP. Further, the SPP studies indicate that ATP and GTP binding can increase the solubility of some proteins.

The results of the experiments are novel especially the demonstration of distance-dependent stabilization and NTP-solubilization of proteins. These solubilization studies may provide insight into protein function which the author hypothesized when discussing the possible role of ATP solubilization of BANF1. The results are of interest to a broad audience including those interested in structure/function relationships as well as those interested in structural information across the proteome. The statistical analysis is rigorous. I recommend the manuscript for publication.

Reviewer #3 (Remarks to the Author):

The manuscript by Sridharan, Zurzawa et al. describes changes to protein stability and solubility upon addition of ATP and GTP. The authors applied several versions of their thermal proteome profiling approach (TPP, 2D-TPP, and SPP) to access the different mechanisms by which NTPs influence the biophysical characteristics of the proteome.

The most remarkable finding is that within the physiological concentration range, ATP modulates the solubility of a significant fraction (25%) of the insoluble proteome. While ATP had been suggested to function as a hydrotrope, to my knowledge this is the first time this function has been assessed at the proteomic scale. In my opinion, the relevance of this finding, along with the elegant experimental design to interrogate phase-transitions proteome-wide, and careful analysis warrants rapid acceptance of the manuscript.

Below are some comments and suggestions that the authors can opt to address for further improving the manuscript:

1. On the initial description of proteins that are affected by ATP (single ATP concentration TPP experiment), it will be interesting to read about additional examples of protein complexes besides the proteasome (p.3, l.90-101).

2. On the NP-40 vs. SDS solubility analysis they authors find the majority of solubilized proteins are annotated as components of membrane-less organelles. Are most of these proteins also annotated as RNA-, DNA-binding proteins?

3. Of the insoluble proteome, ATP can solubilize proteins with high isoelectric point (Fig. 2E). What are the properties and characteristics of the non-solubilized fraction (besides low pI)? Can the authors speculate in the discussion about possible ways (perhaps other small molecules) to solubilize those?

4. There are two in-cell TPP experiments on which ATP is depleted. While this design has many drawbacks (e.g. altered metabolism and associated proteome changes), the experiment adds value because it reproduces their “crude-lysate” findings in cells. The authors acknowledge the limitations and carefully limit the interpretation to validate previous findings without extending further conclusions. I’d suggest moving the text around the limitations from its current location at the description of the second experiment (p.5, l.158-159) to the first time this approach is used (p.4, l.105). Also, if feasible, it will be good to report the ATP concentration in normal vs. 2-deoxyglucose and Antimycin-A treated Jurkat cells?

5. Is there any biological reason for the observed ATP-driven solubilization of proteins? On basis of their results, the authors suggest that metabolic fluctuations can alter the solubility of the proteome. While validating this will require lengthy experiments that are beyond the scope of this work, I suggest the authors search in the literature, if the data is available, for possible correlations between energy state (or ATP concentration) and proteome solubility (or presence, or abundance of membrane-less organelles).

6. For the validation experiment with purified BANF1, it will be best to show the full dose-response curve.

7. p.5, l.171 fix typo BANF > BANF1

We would take this opportunity to thank our reviewers for their time, effort and constructive criticism which has helped us improve the quality of our study. We have included data from additional experiments in response to reviewer 1 and 3's comments. All suggested changes that have been incorporated in the manuscript have been indicated in this letter. Please find our response to the reviewer's comments below in blue:

Reviewer #1 (Remarks to the Author):

In recent years, the non-canonical role of metabolites is gaining increasing importance in cell biology. Building on recent literature, Sridharan et al. present a compelling study for the distinct roles of the molecule ATP on a proteome level by designing a novel method to complement biochemical and proteomic techniques. This study clearly suggests that in addition to the well-known function of ATP as the energy currency of the cell, it can exhibit additional regulatory roles by modulating the solubility of proteins. It is the first to show a general role of ATP in keeping proteins soluble, and the types of proteins that are solubilised by ATP. I strongly recommend the publication of this manuscript after minor improvements.

We thank the reviewer for this very positive and enthusiastic appraisal of our work.

1) In their control experiments of untreated samples authors should use equivalent salt to control for the salt like property of ATP. Or clearly describe what exactly are the conditions in "vehicle treated" controls?

We apologize for the lack of clarity in the describing the experimental conditions used. We have improved the clarity in our methods section, which now explicitly explains the salt concentrations in the vehicle condition. All vehicle conditions contain phosphate buffered saline with 1.5 mM MgCl₂, the medium in which nucleotide triphosphate stocks were prepared. In order to address the question on the influence of Mg-salt on the reported solubility information, we have now included an additional control experiment, where we have compared the fold changes in protein solubility upon addition of 10 mM Mg-ATP and 10 mM MgCl₂ in mechanically disrupted cells (appears now as Supplementary Figure 6E).

The solubility effects observed upon 10 mM Mg-ATP addition differed substantially from those observed in 10 mM MgCl₂ treated samples, showing that ATP induced solubility changes are predominantly not salt driven.

2) The authors always represent the concentration of ATP in the log scale in all the graphs and figures. They should represent the real concentration of ATP to make the figures more coherent with the text and easier to understand.

The figures for which we use log₁₀ concentration of ATP are dose-response curves, these are typically displayed with log₁₀ transformed concentrations.

3) Depending on amount of effort needed, it will be beneficial to be consistent with the use of ATP-Mg and ATP-Na. Or the authors should include a statement on why ATP-Na was used for thermal proteome profiling (TPP) and ATP-Mg for solubility proteome profiling (SPP).

We saw that NaATP can have an effect on some proteins (mostly ribosomal protein) by to some extent depleting endogenous Mg²⁺ levels. While this does not affect protein thermal stability it could affect apparent protein solubility, we therefore switched from NaATP used for 2D-TPP to MgATP used for SPP experiments.

4) In the text the explanation for choosing the candidates in Fig 1 B-C comes after Fig 1E. Authors should try to keep the text and figures chronological.

We thank the reviewer for pointing this out. We now stick strictly to chronological order.

5) In the schematic figures for 2D-TPP and SPP (Fig 1A and 2A) it is not quite clear what the different colors and intensities are representing?

We use shades of purple (increasing intensity) to represent increasing concentration of NTP and shades of grey (increasing intensity) to represent reduction in intracellular ATP. This information is now explained in the figure legend to increase clarity.

6) The authors should pay attention to the numbers that are mentioned in the text are consistent with ones mentioned in the graphs (line 61: 315 ATP binders/ Fig 1D – 313 ATP binders).

We apologize for the inconsistency. We have now fixed the number in the figure. This inconsistency occurred due to the filtering criteria used for generating the figure, which is now harmonized.

7) In a recent paper (J. Kang et al., Biochemical and Biophysical Research Communications 504, 2018), it was pointed out that low concentrations of ATP in micromolar range induce phase separation of the ALS-linked RNA binding protein FUS while millimolar concentrations disrupt phase separation. Will be interesting to observe if this is some general phenomena across several proteins, that might explain a positive feedback of aggregation driven by a decrease in ATP concentration.

We thank the reviewer for bringing an interesting report on ATP-dependent FUS phase transition to our notice. We do not observe global biphasic solubility effects of ATP on proteins, with Nucleolin (NCL) as the only exception (data shown below). However, Nucleolin does not change in solubility upon addition of GTP or AMP-PNP, suggesting that this behavior may be related to the ATP-related metabolism.

Reviewer #2 (Remarks to the Author):

This manuscript uses 2D thermal proteome profiling (TPP) to map NTP-interactions on a proteome-wide scale and solubility proteome profiling (SPP) to examine the role of NTPs in protein solubility. With TPP experiments, the authors were able to detect known ATP and GTP-binding proteins as well identify proteins that are not currently classified as ATP or GTP-binding proteins. The authors also demonstrated that results found in crude lysates could be recapitulated in intact cells. The results also showed that subunits of proteins that do not directly bind ATP or GTP are stabilized when they are in close proximity to subunits that are bound and stabilized by ATP or GTP. Further, the SPP studies indicate that ATP and GTP binding can increase the solubility of some proteins.

The results of the experiments are novel especially the demonstration of distance-dependent stabilization and NTP-solubilization of proteins. These solubilization studies may provide insight into protein function which the author hypothesized when discussing the possible role of ATP solubilization of BANF1. The results are of interest to a broad audience including those interested in structure/function relationships as well as those interested in structural information across the proteome. The statistical analysis is rigorous. I recommend the manuscript for publication.

We thank the reviewer for this very positive appraisal of our work.

Reviewer #3 (Remarks to the Author):

The manuscript by Sridharan, Kurzawa et al. describes changes to protein stability and solubility upon addition of ATP and GTP. The authors applied several versions of their thermal proteome profiling approach (TPP, 2D-TPP, and SPP) to access the different mechanisms by which NTPs influence the biophysical characteristics of the proteome.

The most remarkable finding is that within the physiological concentration range, ATP

modulates the solubility of a significant fraction (25%) of the insoluble proteome. While ATP had been suggested to function as a hydrotrope, to my knowledge this is the first time this function has been assessed at the proteomic scale. In my opinion, the relevance of this finding, along with the elegant experimental design to interrogate phase-transitions proteome-wide, and careful analysis warrants rapid acceptance of the manuscript.

We thank the reviewer for this very positive appraisal of our work.

Below are some comments and suggestions that the authors can opt to address for further improving the manuscript:

1. On the initial description of proteins that are affected by ATP (single ATP concentration TPP experiment), it will be interesting to read about additional examples of protein complexes besides the proteasome (p.3, l.90-101).

We present network diagrams of several additional complexes that exhibit stabilization upon ATP addition along with the Euclidean distances of co-melting behavior of the complex members in supplemental figure S4. However, the lack of information on structural proximity of the different complexes members make it difficult to rationalize the observed propagation of stability. In case of the proteasome the structure is very well studied and we hence could rationalize propagation of stabilization behavior.

2. On the NP-40 vs. SDS solubility analysis they authors find the majority of solubilized proteins are annotated as components of membrane-less organelles. Are most of these proteins also annotated as RNA-, DNA-binding proteins?

Several solubilized proteins that are a part membrane-less organelles are also annotated as RNA-, DNA- binding proteins. The venn diagram (below) illustrates the overlap between different GO:terms among all ATP-solubilized proteins.

3. Of the insoluble proteome, ATP can solubilize proteins with high isoelectric point (Fig. 2E). What are the properties and characteristics of the non-solubilized fraction (besides low pI)? Can

the authors speculate in the discussion about possible ways (perhaps other small molecules) to solubilize those?

Apart from analyzing the extent of disorder and isoelectric points of proteins that solubilized with ATP, we looked at the hydrophobicity of proteins based on their gravity score (see below).

As expected we observe the insoluble proteins are less hydrophobic compared to soluble proteome. There is a mild tendency for ATP-solubilized proteins to have more hydrophobic residues than the insoluble proteins that remain unaffected by ATP, but this trend is not statistically significant. Based on these characteristics, although it may be an overextension, one could speculate that positively charged metabolites, such as e.g. arginine may have an influence on solubility of the proteome. Since we feel that this is still a very speculative prediction, we chose not to bring it up in the manuscript.

4. There are two in-cell TPP experiments on which ATP is depleted. While this design has many drawbacks (e.g. altered metabolism and associated proteome changes), the experiment adds value because it reproduces their “crude-lysate” findings in cells. The authors acknowledge the limitations and carefully limit the interpretation to validate previous findings without extending further conclusions. I’d suggest moving the text around the limitations from its current location at the description of the second experiment (p.5, l.158-159) to the first time this approach is used (p.4, l.105). Also, if feasible, it will be good to report the ATP concentration in normal vs. 2-deoxyglucose and Antimycin-A treated Jurkat cells?

We thank the reviewer for helping us improve the readability of the manuscript. As suggested, we have moved the sentences that acknowledge the drawbacks of ATP depletion to the section where we perform TPP experiments using ATP depleted cells. Since, accurate measurements of absolute intracellular ATP concentrations in cells are challenging, we have reported the relative change in concentration between depleted and not depleted (Fig S5B). This observation is now described in the main text as well.

5. Is there any biological reason for the observed ATP-driven solubilization of proteins? On basis of their results, the authors suggest that metabolic fluctuations can alter the solubility of the proteome. While validating this will require lengthy experiments that are beyond the scope of this work, I suggest the authors search in the literature, if the data is available, for possible correlations between energy state (or ATP concentration) and proteome solubility (or presence, or abundance of membrane-less organelles).

Munder, M. C. *et al* (2016) and Parry, B. R. *et al* (2014) have measured the dynamics of cytoplasmic proteins in yeast and bacteria respectively using single particle tracking techniques during nutrient starvation. These studies observe reduction in the solubility of cytoplasmic proteins during cellular stress. Munder, M. C. *et al*, have further shown that reduction in ATP levels due to nutrient starvation leads to drop in pH of cytosol of yeast which causes a phase transition of cytoplasm from liquid- to a solid- like state. Hence, these observations suggest a correlation of between energy status of a cell and its proteome solubility. We have now included a few sentences referring to these studies in the Discussion.

6. For the validation experiment with purified BANF1, it will be best to show the full dose-response curve.

We thank the reviewer for suggesting this experiment. As suggested, we have now performed the experiment to show the ATP dose-dependency on BANF1 binding to DNA (Fig. 4D – shown below).

We observe a dose-dependent reduction of BANF1 binding, and a 50% reduction around 2.5 mM ATP.

7. p.5, l.171 fix typo BANF > BANF1

We apologize for the typographic error. This is now fixed.

We hope we have been able to address all the questions satisfactorily and include all the suggestions from the reviewers. We would be happy to clarify and answer in case of any further queries.

Thank you.

Yours sincerely,

Marcus Bantscheff and Mikhail Savitski